# Rituximab Use in Warm and Cold Autoimmune Hemolytic Anemia

**DOI:** 10.3390/jcm9124034

**Published:** 2020-12-13

**Authors:** Irina Murakhovskaya

**Affiliations:** Department of Hematology and Oncology, Albert Einstein College of Medicine/Montefiore Medical Center, Bronx, NY 10467, USA; imurakho@montefiore.org

**Keywords:** autoimmune hemolytic anemia treatment, warm autoimmune hemolytic anemia, rituximab, cold agglutinin disease

## Abstract

Autoimmune hemolytic anemia is a rare condition characterized by destruction of red blood cells with and without involvement of complement. It is associated with significant morbidity and mortality. In warm autoimmune hemolytic anemia, less than 50% of patients remain in long-term remission following initial steroid therapy and subsequent therapies are required. Cold agglutinin disease is a clonal hematologic disorder that requires therapy in the majority of patients and responds poorly to steroids and alkylators. Rituximab has a favorable toxicity profile and has demonstrated efficacy in autoimmune hemolytic anemia in first-line as well as relapsed settings. Rituximab is the preferred therapy for steroid refractory warm autoimmune hemolytic anemia (wAIHA) and as part of the first- and second-line treatment of cold agglutinin disease. This article reviews the mechanism of action of rituximab and the current literature on its role in the management of primary and secondary warm autoimmune hemolytic anemia and cold agglutinin disease.

## 1. Introduction

Autoimmune hemolytic anemia (AIHA) is a rare disorder with an estimated incidence of 0.8–3 per 10^5^/year in adults, and prevalence of 17 per 100,000 [1,2,3]. AIHA can present at any age but it is more common in adults with a peak incidence between 50 and 70 years [1]. AIHA is characterized by the production of antibodies directed against red cell antigens with consequent red blood cell destruction with or without the involvement of complement [4].

Autoimmune hemolytic anemia is classified as warm or cold (cold autoimmune hemolytic anemia and paroxysmal cold hemoglobinuria) based on the antibody class and thermal amplitude of the pathogenic antibodies. Warm autoimmune hemolytic anemia (wAIHA) accounts for 80% of the cases of AIHA in adults [5]. The pathogenic antibody is usually immunoglobulin of the IgG subtype, and a direct antiglobulin test (DAT) is positive for presence of IgG and in some cases C3d on the red cell surface, although it can be negative in 5% of cases [4,6]. Destruction of IgG-coated red blood cells (RBC) takes place in the spleen via Fcγ receptor-mediated phagocytosis by splenic macrophages, antibody-dependent cell-mediated cytotoxicity, and complement-mediated hemolysis [7]. 

In cold AIHA, the pathogenic antibody is a cold agglutinin most often of the immunoglobulin M (IgM) subtype with I antigen specificity. Cold agglutinin fixes complement (C1q) below core body temperature and dissociates from the red cells as they move centrally, which leads to activation of the classical complement pathway and deposition of C3b on the RBC surface [8]. DAT is positive for presence of C3d. Complement-coated erythrocytes are subject to extravascular hemolysis via C3b receptors in the hepatic mononuclear phagocytic system. If a large amount of complement is generated, formation of the membrane attack complex leads to intravascular hemolysis. This process is generally limited by the presence of complement regulatory proteins (CD55 and CD59) on the red blood cell surface. In some cases of AIHA, both pathogenic IgG antibody and clinically significant cold agglutinin are present. Those are referred to as “mixed” AIHA. Paroxysmal cold hemoglobinuria is characterized by a biphasic cold-reacting IgG, which binds to patient RBCs in the cold; fixes complement; and dissociates at body temperature, causing hemolysis [9,10].

In approximately 50% of cases, AIHA is associated with an underlying collagen vascular disease, lymphoproliferative disorder, infection, solid tumors, or medication, and in the other 50% it is idiopathic [11]. Autologous and allogeneic transplants [12,13,14], and immune checkpoint inhibitor therapy for solid tumors [15,16], are emerging etiologies for AIHA.

A clinical course can be variable and severe with a reported mortality rate of 11% [17]. The degree of anemia depends on antibody characteristics, the activity of the mononuclear phagocytic system, and the bone marrow compensatory response [1,18]. Patients can experience acute or chronic hemolytic anemia; hemoglobinuria can be seen with significant complement involvement and intravascular hemolysis. The risk of thrombosis is increased in warm AIHA, particularly during the active phase of the disease [17,19], as well as in cold agglutinin disease [20,21]. In cold agglutinin disease, cold-induced intravascular agglutination results in circulatory symptoms such as acrocyanosis and/or the Raynaud phenomena, which occur in 50–90% of patients [22,23].

## 2. AIHA Pathophysiology

Multiple mechanisms are involved in the pathogenesis of AIHA. In warm autoimmune hemolytic anemia, B cell mediated immune dysregulation results in production of pathogenic autoantibodies, cytokines, as well as T cell activation [4,24]. In addition, T cell dysregulation results in loss of immune tolerance. Decreased levels of regulatory T cells (Tregs), which play a key role in the immunologic self-tolerance, are associated with autoreactivity in AIHA [25,26]. Altered Th1/Th2 balance in favor of Th1 lymphocytes has been demonstrated in patients with autoimmune hemolytic anemia. [27]. Th17 cells, a subset of CD4+ T helper cells which secrete interleukin (IL)-17, have also been implicated in pathogenesis of AIHA and other autoimmune diseases [28]. Increased IL-17 secretion has been shown to be correlated with the disease activity in AIHA patients. [29]. Increased ratio of T follicular helper cells to T follicular regulatory cells, which are essential in the regulation of germinal center reactions and antibody production, were shown to play important role for control and induction of AIHA [30]. 

Unlike warm autoimmune hemolytic anemia, cold agglutinin disease (CAD) is a clonal hematologic disorder. Primary CAD is a distinct low grade lymphoproliferave disorder [31,32,33], with discernible histologic [31] and flow cytometric characteristics [32], immunoglobulin heavy and light chain gene features [33], cytogenetic abnormalities [34], and recurrent somatic mutations on next generation sequencing [35]. The pathogenic IgM is monoclonal, with kappa light chain specificity in 85% of cases [22,36,37]. Secondary cold agglutinin syndrome associated with B cell lymphoproliferative disorders is less common than primary CAD. In infection associated cAIHA, which is usually transient, IgM is polyclonal [38].

## 3. Rituximab Mechanism of Action

Rituximab is a chimeric anti-CD20 antibody that consists of human IgG1-kappa constant region and variable regions from the murine monoclonal anti-CD20 antibody [39]. The B-cell antigen CD20 is a transmembrane protein that is present in virtually all B cells from the stage of commitment to B-cell lineage to differentiation into plasma cells, at which point it is downregulated. [40]. 

Rituximab was initially developed as a therapy for CD20+ lymphoproliferative malignancies. Subsequently, efficacy was also demonstrated in autoimmune disorders such as rheumatoid arthritis (RA); systemic lupus erythematosus; multiple sclerosis; and several hematologic autoimmune diseases such as immune thrombocytopenic purpura (ITP), acquired hemophilia, thrombotic thrombocytopenic purpura, and AIHA [41,42,43,44,45]. In AIHA, rituximab demonstrated efficacy in the first line as well as relapsed setting [43,46,47,48,49,50,51,52] in primary as well as secondary forms [49,53] of wAHA, cold agglutinin disease, and secondary cold agglutinin syndrome.

Rituximab binding to CD20 on the B cell surface induces B cell apoptosis via direct signaling, activation of complement, and engagement of the cell-mediated cytotoxicity [39,54]. The resultant B cell depletion, which can persist for 6-12 months, [40,55] leads to suppression of B-cell mediated production of auto-Abs, cytokine secretion, and their APC function [56]. In addition to B cell depletion, rituximab has an immunomodulatory effect on cellular and innate immunity in autoimmune diseases. ITP therapy with rituximab has been associated with upregulation of the T reg cells and restoration of Th1/Th2 ratio [57,58,59]. In nephrotic syndrome, rituximab therapy was associated with decrease in Th17, Th2, and memory T cells, and increased Treg cells [60]. Clinical response to rituximab therapy in AIHA was found to be associated with decrease in Th1 cytokines interferon-γ, interleukin-12, tumor necrosis factor-α, and IL-17, suggesting immunomodulatory activity of rituximab on Th1/Th2 balance [61].

## 4. Treatment of Warm Autoimmune Hemolytic Anemia

Management of wAIHA has historically been empirically derived or based on retrospective studies with paucity of randomized and prospective clinical trials [1,62]. Steroids have been used as a first line therapy with initial response rate of 70–80% [17,50,63]. Up to 20-40% of patients have a durable remission after initial therapy, but the rest have a chronic, relapsing course [2,17,64], requiring subsequent therapy [50]. 

Splenectomy has been used traditionally as a second-line therapy for wAIHA with initial response rates of 60–70% [65,66,67], and surgical mortality rate of 0.8% in recent studies [68]. However, long-term cure rates are only approximately 20% [2] with associated risk of infections due to encapsulated bacteria of 3.3–5% [69,70]. Thrombotic complications [71] and pulmonary hypertension [72] are also increased in splenectomized patients. More recently, Rituximab has emerged as an effective therapy of wAIHA, replacing splenectomy as preferred modality in the second-line setting [7,73,74,75,76,77].

Additional therapeutic options in wAIHA include immunosuppressive agents. Azathioprine and cyclophosphamide have been used in relapsed setting with overall responses reported in one-third of patients [2,78]. These response rates are based on the retrospective studies and case reports and are subject to selection and publication bias. Responses to therapy with other immunosuppressive agents such as cyclosporine and mycophenolate mofetil have been reported as well, although data are limited [2].

Danazol was reported to produce responses both as a single agent and in combination with prednisone [79] in retrospective studies. Administration of erythropoietin has been successfully used in patients with refractory AIHA, particularly in the presence of reticulocytopenia [18]. Several novel agents are currently under investigation, including Syk inhibitor Fostamatinib [80], FcRn receptor inhibitors [81,82], and Bruton tyrosine kinase (BTK) inhibitors [83,84]. Figure 1 summarizes current therapeutic algorithm for wAIHA.

## 5. Rituximab in Warm Autoimmune Hemolytic Anemia

### 5.1. Relapsed/Refractory wAIHA

In a small prospective pediatric study in 15 children with relapsed/refractory wAIHA, rituximab therapy was associated with an 87% response rate [85]. Several retrospective observational studies reported efficacy of rituximab in treatment of relapsed/refractory wAIHA. The largest studies that included primary wAIHA are summarized in Table 1. Rituximab demonstrated efficacy in patients with both primary and secondary AIHA with overall response rates (ORR) of 70–80% in relapsed/refractory setting and a median duration of response of 1–2 years. Median time to response was 4 to 6 weeks after the first rituximab dose, with majority of patients responding within 4 weeks after the first dose [48,86]. Responses within the first week [49] and up to 3 months [86] were also seen. Complete response (CR) rates were more variable, ranging from 25 to 75%, reflecting clinical heterogeneity of the disease. Relapse rate at 1–2 years was 25–50% [43,86], with retreatment associated with responses [49,85,86].

A meta-analysis of 21 observational studies that included adults or children with primary or secondary cold and warm AIHA treated with rituximab reported similar results, with overall response rate (ORR) of 79% and CR rate of 42% for wAIHA. CR rate was the highest within 2 to 4 months after rituximab initiation. Younger patients, and shorter duration of wAIHA, had a better response to rituximab treatment [87].

### 5.2. Rituximab as First Line Therapy of wAIHA 

Two prospective randomized phase III trials [88,89] evaluated role of rituximab in the front-line setting. A Danish open-label randomized study compared rituximab with prednisolone with prednisolone monotherapy in 64 newly diagnosed primary wAIHA patients. Conventional lymphoma schedule of 4 weekly intravenous rituximab 375 mg/m^2^ infusions was administered. Addition of rituximab to prednisolone resulted in significantly higher response rates at 12 months (75% vs. 36%) and improved relapse free survival at 36 months (70% vs. 45%) compared to prednisolone alone [88]. 

A more recent randomized double-blind placebo-controlled trial in 32 adults with newly diagnosed primary wAIHA demonstrated similar results with higher efficacy (ORR 75%/34% CR) with rituximab compared to placebo (ORR 31%/16% CR). Two infusions of rituximab or placebo at a fixed dose of 1000 mg were administered 2 weeks apart. Patients in both arms received prednisone 1 mg/kg day in the first 2 weeks [89]. Response duration was longer with rituximab compared to placebo, with 10/16 patients treated with rituximab and 3/16 patients in the placebo arm remaining in CR at 2 years. 

### 5.3. Rituximab Combination Therapy

Rituximab combination therapy with cyclophosphamide and dexamethasone (RCD) demonstrated significant efficacy in chronic lymphocytic leukemia (CLL)-associated wAIHA [90,91]. This combination has also been evaluated in primary wAIHA. Response rate of 97% with CR 42% was reported by Bocian et al. in 19 wAIHA cases (12 primary AIHA and 6 treatment naïve) treated with RCD in the first line and relapsed setting [92]. Similar response rates were seen in 16 relapsed/refractory wAIHA with OR 94% and CR of 69% [93]. Among patients who achieved CRs, majority (64%) had secondary WAIHA. The median duration of response was 9.8 months. Two patients required hospitalization for infection. Taking into consideration single agent rituximab activity in wAIHA, further studies are needed to evaluate the role of RCD in wAIHA therapy.

### 5.4. Rituximab in Secondary wAIHA

Secondary wAIHA is most commonly seen in association with hematological malignancies, infection, immunodeficiency, and autoimmune disorders [50]. A general principle for therapy of secondary wAIHA is if the underlying disorder is clinically active, initial therapy should include treatment of underlying cause [74,77]. If underlying disease is adequately controlled, treatment of secondary wAIHA is generally similar to that of primary wAIHA. 

Rituximab has shown efficacy in relapsed setting in CLL as a single agent [53] and in combination with cyclophosphamide and dexamethasone (RCD) [90,91]; cyclophosphamide, vincristine, and prednisone (CVP) [94]; and with bendamustine [95] with response rates of 80–100% and response duration of 22–24 months with combination therapy (Table 2). In AIHA associated with systemic lupus erythematosus (SLE) [96] and common variable immune deficiency (CVID) [97], responses are approximately 80%, with majority being complete responses. 

AIHA associated with non-CLL lymphoproliferative disorders has lower response rate to steroids, higher rates of steroid dependency, and higher likelihood to require second-line therapy [50,51]. Despite limited data, rituximab containing lymphoma-directed regimens are associated with higher probability of response [51]. Similarly, in AIHA associated with transplantation, responses to steroids are poor and front-line rituximab therapy is associated with better outcomes [98,99].

## 6. Treatment of Cold Agglutinin Disease

Corticosteroids have limited efficacy in treating CAD with response rates of 14–35% [17,22], and need high doses to maintain remission [1,38,100]. Cytototoxic immunosuppressive drugs such as chlorambucil or cyclophosphamide have been tried with limited efficacy, with partial responses seen in 16% patients in retrospective studies [22,101]. Splenectomy is not effective, since clearance of C3b-opsonized red blood cells occurs primarily in the liver [1,22].

Single agent rituximab as well as rituximab containing immunochemotherapy have significant activity and are currently recommended as first- and second-line therapy [78,102] in CAD. Bortezomib has shown promising activity in a small prospective, non-randomized study of 19 patients with CAD. Responses were seen in 32% patients after a single cycle of therapy, with 16% complete responses [103]. 

Complement inhibition with antibody directed therapy against C5 with eculizumab [104], C3 with pegcetacoplan [105], and most recently C1s sutimlimab [106] has shown efficacy and is further being explored in clinical trials [107,108]. In the phase 1b, first-in-human trial in patients with CAD treated with four weekly doses of sutimlimab, 7 out of 10 patients responded with a hemoglobin increase >2 g/dL. Responses were rapid with median increase in hemoglobin levels of 1.6 g/dL within the first week, and 3.9 g/dL within 6 weeks. All previously transfused patients achieved transfusion independence [109]. These responses were recapped in a named patient program in responders with every 2 weeks maintenance resulting in sustained increase in hemoglobin to near normal levels and transfusion independence in all patients upon re-exposure [110]. Sutimlimab was granted breakthrough therapy designation by the US Food and Drug Administration for the treatment of CAD based on these data and a Phase 3 trial recently completed the accrual.

Phosphoinositide 3-kinase (PI3K) inhibitors [111] are in trials as well, and Bruton’s tyrosine kinase (BTK) inhibitor ibrutinib has also shown promising results [112]. Figure 2 summarizes current treatment algorithm for cold agglutinin disease.

### Rituximab in Primary CAD

Several retrospective studies reported efficacy of rituximab in cold autoimmune hemolytic anemia, including secondary cold agglutinin syndrome and primary cold agglutinin disease with response rates of 60% [43,49]. In two prospective studies in CAD, single-agent rituximab therapy was associated with overall response rate of ~50% [113,114]. Complete responses were rare, and median duration of response was short at only 6–11 months.

Recognition of presence of an underlying lymphoproliferative disorder in CAD prompted consideration for incorporation of the cytotoxic-lymphoma-directed therapy in rituximab-based therapy. A Norwegian multicenter trial prospectively evaluated a combination of rituximab and fludarabine in 29 patients with CAD. Four 28-day cycles of oral fludarabine 40 mg/m^2^ on days 1–5 with standard-dose rituximab 375mg/m^2^ were administered. Responses were seen in 76% of patients with complete response rate of 21%, including patients previously treated with rituximab. Estimated median response duration was greater than 66 months. Toxicity was considerable, with 41% grade 3–4 hematologic toxicity; 59% grade 1–3 infections, including recurrent infections; and 10% herpes zoster reactivation [115]. A 10-year update of this study reported ORR 62% and CR 38%, with median estimated response duration of 77 months [23]. The rate of late onset second malignancies was 31%, which was numerically higher than all CAD patients and those treated with rituximab/bendamustine combination (13% and 9%, respectively), although it did not reach statistical significance.

A more recent study investigated combination of rituximab with bendamustine in a prospective, nonrandomized multicenter trial of 45 patients with CAD. Rituximab 375 mg/m^2^ day 1 and bendamustine 90 mg/m^2^ days 1 and 2 were administered for 4 28-day cycles. Overall response rate was 71% with 40% complete response. Responses were observed in 50% of cases previously treated with rituximab or rituximab-fludarabine therapy. Acrocyanosis and Raynaud symptoms resolved completely in 50% of patients and improved in additional 32%. Grade 3–4 neutropenia was seen in 33% of patients; 11% of patients experienced infections, and 29% of patients required hospitalization for a median of 8 days [116]. In a recent update [23], the authors reported improved response rates with ORR 78% and CR of 53%, suggesting that responses deepen over time. This was attributed to the prolonged time to response seen in many patients. In patients responding to rituximab-bendamustine, median response duration was not reached after 88 months, and estimated 5-year sustained remission rate was 77%. A combination of rituximab with or without bendamustine is currently recommended in the first line for patients with cold agglutinin disease requiring therapy and in second-line therapy [77]. 

## 7. Rituximab Dose

Most studies of rituximab in autoimmune hemolytic anemia utilized conventional dose of 375mg/m^2^ weekly for 4 weeks used in treatment of lymphoproliferative diseases. Recognition that lymphocyte burden in autoimmune diseases is lower than the tumor mass in lymphoproliferative disorders led to consideration of lower dose rituximab with potential benefit of lower cost and toxicity. Lower doses of rituximab (100 mg fixed dose) have been studied in various autoimmune settings such as rheumatoid arthritis, vasculitis, thrombotic thrombocytopenic purpura, and immune thrombocytopenic purpura [57,117,118,119].

In steroid refractory patients with autoimmune cytopenia, low dose rituximab in combination with anti-CD52 monoclonal antibody alemtuzumab produced responses in 100% of patients and complete response in 58% [120]. In wAIHA, a retrospective analysis of 64 adults reported response rates of 76.7% to steroids alone and 100% to rituximab at 100 mg/week for 4 weeks in combination with high-dose dexamethasone [52]. 

A prospective Italian multicenter study of low-dose rituximab in warm primary AIHA and CAD demonstrated responses comparable to standard-dose rituximab that were sustained [61,121]. In wAIHA, overall response rate was 80–100% with CR of 60–80% at 12 months. Response rate in CAD was inferior at around 50%, with higher relapse rates. Ten-year follow up of this data [122] confirmed the efficacy of lower dose rituximab in primary AIHA. Overall median duration of response was 1 year and 3 months, with 62% of the patients relapsing. In wAIHA, responses deepened overtime, supporting postulated immunomodulatory effect of rituximab. Relapse-free survival (RFS) was significantly longer in wAIHA than in other forms (mixed, atypical forms and CAD) at 64 months vs. 25 months and correlated with the depth of the response. Responses were seen with retreatment with both conventional-dose and low-dose rituximab. 

## 8. Rituximab Toxicity

Rituximab has been used extensively in hematologic malignancies, where it has demonstrated a well-established safety profile [123]. Approximately 40% of patients experience mild to moderate infusion-related reactions, mostly fever and chills, during the first administration of rituximab [87]. The incidence of infusion reactions decreases markedly with subsequent infusions. Rheumatoid arthritis patients receiving rituximab exhibited adverse effect profile similar to patients with lymhoproliferative disorders, but the overall incidence and severity were considerably lower, possibly due to lack of cytokine release associated with lysis of tumor cells. Neutropenia is seen in 2% patients and hypogammaglobinemia is seen occasionally, but infections are not common [124].

Rare cases of serious viral infections such as John Cunningham polyomavirus (JCV) reactivation leading to progressive multifocal encephalopathy (PML), hepatitis B reactivation, as well as other infections have been reported [124,125,126,127,128,129]. Majority (65%) of PML cases are diagnosed in the first 2 years after the first rituximab dose (median time 16 months) and are fatal in 90% of cases [125]. In a recent review of PML cases reported in a FAERS and EudraVigilance databases, approximate incidence rate of PML ranged from 1.39 to 1.87 per 10,000 exposed patients [126]. Potential synergies, particularly with the purine analog fludarabine, alkylating agent bendamustine, and hematopoietic stem cell transplantation, have been observed, with higher incidence rates of PML in CLL and Non- Non-Hodgkin’s Lymphoma (NHL) compared to RA patients. 

Association of rituximab therapy with hepatitis B viral reactivation was first observed in lymphoma patients [127]. In rheumatoid arthritis, the incidence of reactivation was reported to be 9.1% among the hepatitis B surface antigen negative/hepatitis B core antibody positive ( HBsAg−/anti-HBc+ ) patients with mean duration of 25.4 months from the first rituximab dose [128]. Pre-treatment screening with hepatitis B serologies and prophylactic antiviral therapy for carriers is recommended in patients who are treated with anti-CD20 therapy [77].

A meta-analysis of 21 studies comprising 154 patients with AIHA treated with rituximab reported toxicities in 14% patients. Of those, infusion-related side effects, mostly fever and chills, were seen in 42%. The remaining 58% were more severe, with 2.6% neutropenias, 12% severe infections, and Pneumocystis jiroveci pneumonia in one patient [87].

## 9. Discussion

Autoimmune hemolytic anemia is a diverse group of diseases with substantial clinical heterogeneity. Rituximab therapy demonstrated significant efficacy in autoimmune hemolytic anemia in primary as well as secondary forms, and in frontline as well as relapsed settings. In wAIHA, rituximab plays an established role in relapsed disease, providing a safe and effective alternative to splenectomy. Therapy with rituximab in the first line setting is associated with higher response rates and improved relapse-free survival compared to steroids and allows for limited treatment duration. Cost of rituximab therapy is significant compared to glucocorticoid therapy. Front line rituximab therapy can be considered in selected cases where prolonged corticosteroid therapy is not desirable. 

In cold agglutinin disease, rituximab alone or in combination with bendamustine is effective in alleviating anemia as well as cold-induced circulatory symptoms and is currently the recommended therapy in the first as well as second-line settings [77]. Responses can take up to 6 weeks, and infectious complications are a concern with chemotherapy-based regimens. Further studies are needed to investigate the role of combination of rituximab with novel B cell directed therapies for potential chemotherapy-free combinations. 

Use of low-fixed-dose rituximab in wAIHA demonstrated comparable efficacy to standard-dose rituximab and can ameliorate cost as well as toxicity considerations. Prospective studies comparing various dose levels are needed. Patients with cold agglutinin disease have higher lymphocyte burden in the setting of clonal lymphoproliferative disorder and derive more benefits from conventional dose rituximab [17].

Although relapses are common, retreatment with rituximab is generally effective in reestablishing remission. Rituximab maintenance therapy has not been prospectively studied in autoimmune hemolytic anemia and is a potential direction for further investigation, particularly in multiple-relapse patients.

Several novel agents targeting various aspects of pathophysiology of autoimmune hemolytic anemia are currently under investigation for wAIHA and CAD, expanding therapeutic options for these uncommon and challenging to treat conditions. Sequencing these therapies and combining them with rituximab are potential challenges to be addressed in future clinical trials.

## Figures and Tables

**Figure 1 jcm-09-04034-f001:**
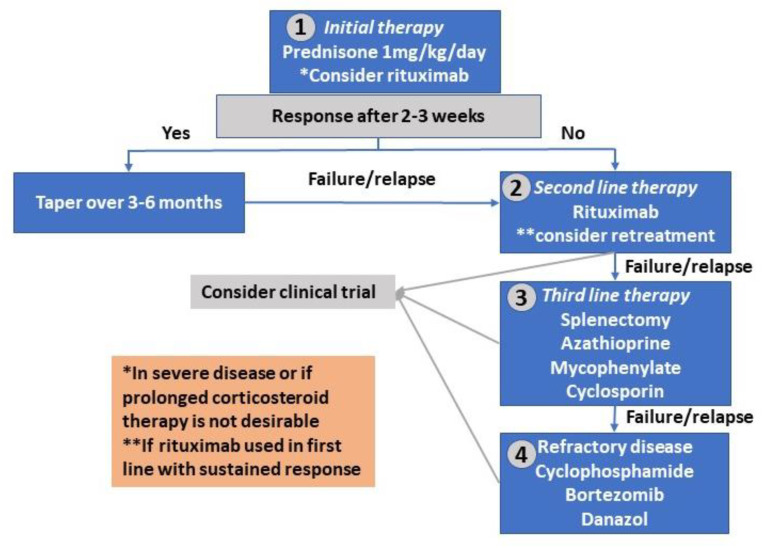
Treatment algorithm for warm autoimmune hemolytic anemia. “*” = in severe disease or if prolonged corticosteroid therapy is not desirable; “**” = if rituximab is used in the first line with sustained response.

**Figure 2 jcm-09-04034-f002:**
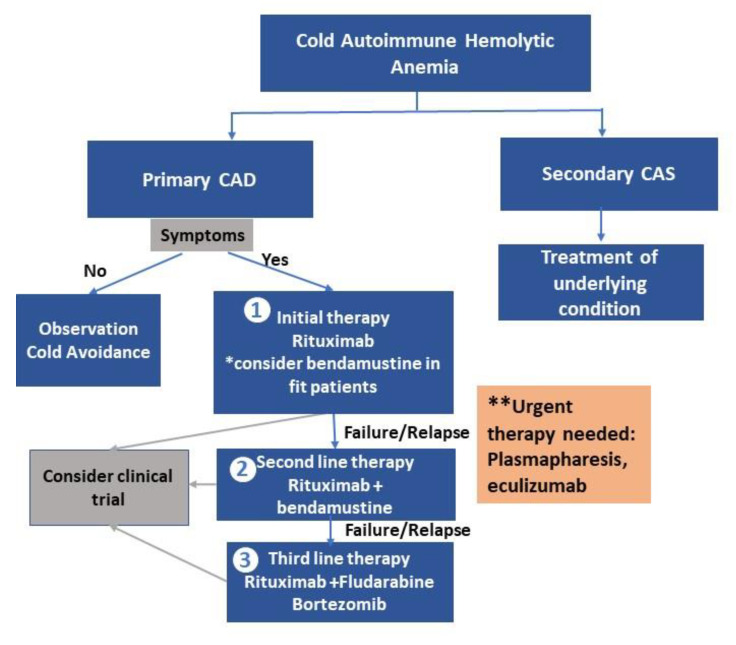
Treatment algorithm for cold agglutinin disease. CAD = cold agglutinin disease; CAS = cold agglutinin syndrome; “*” = consider bendamustine in fit patients; “**” = cases when urgent therapy is needed.

**Table 1 jcm-09-04034-t001:** Retrospective trials of rituximab in relapsed/refractory warm autoimmune hemolytic anemia (wAIHA).

First Author (Year)	Number of Patients	Splenectomy (N)	ORR/CR (%)	Duration of Response (mo)	Comment
Narat (2005) [46]	11	5	64/27	2–20 (median 11)	
D’Arena (2007) [47]	11	1	100/73	1–96+	All primary AIHA, 2 patients received 3 additional monthly doses of maintenance R; additional patient had Rituxan retreatment for ITP
Bussone (2009) [48]	27	6	93/30	NR	5 (18% relapses) after median f/u 20.9 mo, 3 retreated with R and responded
Dierickx (2009) [43]	36	10	83/50	1yr PFS 72% 2yr PFS 56%	
Peñalver (2010) [49]	27	13	77/61	Duration of response > 6mo in patients in CR	
Maung (2013) [86]	34	3	71/27	9–60	50% relapse; median time to next treatment 16.5mo 28.5% maintained response at 3 years
Roumier (2014) [50]	25	2	80/NR	50% relapse after a mean of 14 ± 8	62% secondary wAIHA
Barcellini (2014) [17]	32	NR	81/56		Primary AIHA only low dose R
Jaime-Pérez (2019) [52] First-line Relapsed	18 8	N/A 7	100/83 100/63	Median 16.5 Mean maintained response 82 ± 18	Low dose R + high dose dexamethasone (40 mg/day) for 4 days

R = rituximab; low dose rituximab: rituximab at 100 mg/week for 4 weeks; NR = not reported; ORR = overall response rate; CR = complete response; PFS—progression free survival; ITP = immune thrombocytopenic purpura; N/A = not applicable; mo = months.

**Table 2 jcm-09-04034-t002:** Rituximab in secondary wAIHA.

Underlying Condition	First Author (Year)	Regimen	Number of Patients	ORR/CR (%)	Duration of Response (mo)	Line of Therapy
CLL	D’Arena (2006) [53]	R 375 mg/m^2^ × 4	14	72/22	NR	R/R
CLL	Rossignol (2011) [90]	rituximab, cyclophosphamide, and dexamethasone (RCD)	26	89.5/81	24	R/R (including R-refractory
CLL	Kaufman (2009) [91]	RCD	7 13	100/85	22	1st line R/R
CLL	Bowen (2010) [94]	R-CVP	17	100/82	21.7	R/R (including R-refractory
CLL	Quinquenel (2015) [95]	R-bendamustine	25 1	81/31	28	1st line R/R
non-CLL LPD	Hauswirth (2007) [51]	R, R-chemo	7	100/100	NR	R/R
SLE	Serris (2018) [96]	R, R-immunosuppression	16	87.5/75	RFS 62.5% at 2 years	R/R
CVID	Gobert (2011) [97]	R	5 AIHA 5 AIHA/ITP	80/80 80/40	NR	R/R
Transplant associated	Faraci (2014) [99]	R, immunosuppression	7	100/100	NR	R/R
Transplant associated	Sanz (2014) [98]	R, immunosuppression	4 4	75/50% 50%/0	NR	1st line R/R

R = rituximab; R-CVP-rituximab, cyclophosphamide, vincristine, and prednisone; R/R-Relapsed/Refractory. NR = not reported; ORR = overall response rate. CR = complete response. RFS—relapse free survival. ITP = immune thrombocytopenic purpura; AIHA = autoimmune hemolytic anemia; wAIHA = warm autoimmune hemolytic anemia; mo = months.

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
