# Peer review of "Rituximab Use in Warm and Cold Autoimmune Hemolytic Anemia"

_jcm, 2020, doi:10.3390/jcm9124034_

Round 1
Reviewer 1 Report
In this paper, Murakhovskaya reviews the literature and discusses the use of rituximab in warm- and cold-antibody mediated autoimmune hemolytic anemia. The selection of references is appropriate, comprehensive, and impressively up-to-date. The conclusions are clearly based on the findings (literature reviewed), and the article is well-written. If published, this will serve as a comprehensive review on the theoretical aspects of rituximab-containing therapies in these patients as well as an updated outline on the clinical use. I have some suggestions that, if followed, might further improve the high quality of this article.
Major comment
- The article would profit from a slightly more comprehensive and detailed, but still concise Discussion in which the author provides her qualified opinion based on the preceding literature review. The existing Discussion is very brief, more like a Conclusion, and leaves some obvious questions unanswered. For example, section 7 (Rituximab dose) provides an excellent review of the literature and identification of different response rates in wAIHA versus CAD, but the practical consequence of this difference should be briefly mentioned in the Discussion.
Minor scientific comments
- Line 36. The phrase “IgM is a cold agglutinin…” should be rewritten because warm-reactive IgM autoantibodies also do exist in the context of AIHA.
- Line 50. Explain “mortality rate”. One-year, 5-year, 10-year mortality rate, etc.?
- Line 116-118. “Azathioprine and…”: These response rates are based on an old retrospective study (Author’s reference #78) and some pooled case reports. The latter type of analysis must be interpreted with skepticism because of selection bias, publication bias, and non-defined or heterogeneous response criteria. Thus, although the response rate of 1/3 seems reasonable in my experience, these reservations should be briefly mentioned.
- Line 174-5. Other autoimmune diseases associated with secondary wAIHA are not always etiologies. For example, some coexisting autoimmune diseases (ITP, autoimmune hepatitis, etc. etc.) should be considered as associations (different manifestations of a common cause) rather than causes. I would avoid the term ‘etiology’ in this context.
Suggestions for minor edits
Line 34. ADCC: Although this acronym may be familiar to most readers, is should be explained.
Line 40 (and other occurrences): Consider using the more recent term “mononuclear phagocytic system” instead of “reticuloendothelial system”.
Line 43-44. The term “paroxysmal cold hemoglobinuria” should not be capitalized.
Line 98 and other occurrences: The generic name rituximab should begin with r, not R.
Line 152-3: The corticosteroid studied in this trial (Ref. #88) was prednisolone, not prednisone
Line 202: The rather small number of patients in this trial (Ref. #103) does not permit the use of decimals in the percentages (32%, not 31.6%).
Figure 2: I don’t see from the algorithm what the two asterisks in the separate textbox denote. Please add some symbol etc. to explain. Otherwise an excellent treatment algorithm!
Line 304-5. “A potentially chemotherapy free combinations”: Please check grammar.
Author Response
Response to Reviewer 1 Comments
Point 1:
Major comment
The article would profit from a slightly more comprehensive and detailed, but still concise Discussion in which the author provides her qualified opinion based on the preceding literature review. The existing Discussion is very brief, more like a Conclusion, and leaves some obvious questions unanswered. For example, section 7 (Rituximab dose) provides an excellent review of the literature and identification of different response rates in wAIHA versus CAD, but the practical consequence of this difference should be briefly mentioned in the Discussion.
Response 1:
I would like to thank Reviewer 1 for careful revision and for the thoughtful remarks. As suggested, discussion session has been expanded including elaboration on the practical implications of differences in response rates to low dose rituximab in wAIHA and CAD.
Point 2:
Minor scientific comments
- Line 36. The phrase “IgM is a cold agglutinin…” should be rewritten because warm-reactive IgM autoantibodies also do exist in the context of AIHA.
- Line 50. Explain “mortality rate”. One-year, 5-year, 10-year mortality rate, etc.?
- Line 116-118. “Azathioprine and…”: These response rates are based on an old retrospective study (Author’s reference #78) and some pooled case reports. The latter type of analysis must be interpreted with skepticism because of selection bias, publication bias, and non-defined or heterogeneous response criteria. Thus, although the response rate of 1/3 seems reasonable in my experience, these reservations should be briefly mentioned.
- Line 174-5. Other autoimmune diseases associated with secondary wAIHA are not always etiologies. For example, some coexisting autoimmune diseases (ITP, autoimmune hepatitis, etc. etc.) should be considered as associations (different manifestations of a common cause) rather than causes. I would avoid the term ‘etiology’ in this context.
Response 2:
- I would like to thank the Reviewer 1 for this valid point and the text has been revised accordingly
- Mortality rate reported in the studies has not been well defined. In the study cited median duration of follow up was 33 months.
- As suggested, limitations of retrospective nature of the studies has been discussed. I added the following sentence: “These response rates are based on the retrospective studies and case reports and are subject to selection and publication bias.”
- I appreciate Reviewer 1 comment. The language has been changed as follows: “Secondary wAIHA is seen in association with hematological malignancies, infection, immunodeficiency, and autoimmune disorders.”
Point 3:
Suggestions for minor edits
Line 34. ADCC: Although this acronym may be familiar to most readers, is should be explained.
Line 40 (and other occurrences): Consider using the more recent term “mononuclear phagocytic system” instead of “reticuloendothelial system”.
Line 43-44. The term “paroxysmal cold hemoglobinuria” should not be capitalized.
Line 98 and other occurrences: The generic name rituximab should begin with r, not R.
Line 152-3: The corticosteroid studied in this trial (Ref. #88) was prednisolone, not prednisone
Line 202: The rather small number of patients in this trial (Ref. #103) does not permit the use of decimals in the percentages (32%, not 31.6%).
Figure 2: I don’t see from the algorithm what the two asterisks in the separate textbox denote. Please add some symbol etc. to explain. Otherwise an excellent treatment algorithm!
Line 304-5. “A potentially chemotherapy free combinations”: Please check grammar.
Response 3:
I would like to thank the Reviewer 1 for these helpful remarks
Minor edits have been accepted and corrected

Reviewer 2 Report
After a concise update of the complex pathophysiology of Autoimmune Haemolytic Anaemias and the Rituximab mechanism of action, this article reviews the role of the drug in the context of the multiple, diverse options for the treatment of this heterogeneous group of immune haemolytic disorders.
The review is comprehensive and highly informative, relevant for clinical haematology. There are no general or specific comments, just a few minor points.
References -Two recent papers on the promising complement C1s inhibitor could be cited:
- Gelbenegger G et al. Inhibition of Complement C1s in patients with cold agglutinin disease: a lesson learned from a named patient program. Blood Adv 2020;4:997-1005 .
- Jager U et al. Inhibition of complement C1s improve severe haemolytic anaemia in cold agglutinin disease: a first-in-human trial. Blood 2019;133: 893-901.
Table 2 - In the last column the definition of therapy lines should be made uniform-
Fig. 2 – The first citation of Fig.2 at line 201 can be omitted, as the figure is already more appropriately cited on line 207.
Typos - Lines 17, 40, 41, 54 and 292 should be re-read as they contain some printing errors.
Author Response
Response to Reviewer 2 Comments
After a concise update of the complex pathophysiology of Autoimmune Haemolytic Anaemias and the Rituximab mechanism of action, this article reviews the role of the drug in the context of the multiple, diverse options for the treatment of this heterogeneous group of immune haemolytic disorders.
The review is comprehensive and highly informative, relevant for clinical haematology. There are no general or specific comments, just a few minor points.
Point 1:
References -Two recent papers on the promising complement C1s inhibitor could be cited:
- Gelbenegger G et al. Inhibition of Complement C1s in patients with cold agglutinin disease: a lesson learned from a named patient program. Blood Adv 2020;4:997-1005.
- Jager U et al. Inhibition of complement C1s improve severe haemolytic anaemia in cold agglutinin disease: a first-in-human trial. Blood 2019;133: 893-901.
Response 1:
I would like to thank Reviewer 2 for the kind remarks.
References: The section on therapy of cold agglutinin disease has been expanded to include the discussion on the C1s inhibitor sutimlimab and suggested references incorporated.
Point 2:
Table 2 - In the last column the definition of therapy lines should be made uniform-
Response 2:
I would like to thank the Reviewer 2 for pointing this out.
Table 2 has been modified to make the definition of therapy uniform.
Point 3:
Fig. 2 – The first citation of Fig.2 at line 201 can be omitted, as the figure is already more appropriately cited on line 207.
Response 3:
Fig. 2 The first citation was removed
Point 4:
Typos - Lines 17, 40, 41, 54 and 292 should be re-read as they contain some printing errors.
Response 4:
I appreciate Reviewer 2 pointing out these printing errors. Typos have been corrected.
